# The Role of IRF8 Polymorphisms in Systemic Sclerosis Development and Pathogenesis

Anna Mennella [1,†], Giuseppe Ocone [1,†], Katia Stefanantoni [2] and Loredana Frasca [1,*]

1   National Center for Global Health, National Institute of Health, 00161 Rome, Italy;
    anna.mennella@iss.it (A.M.); giuseppe.ocone@iss.it (G.O.)
2   Department of Clinical and Internistic Sciences, Anesthesiology and Cardiovascular Sciences, University of
    Rome, 00185 Rome, Italy; katia.stefanantoni@uniroma1.it
*   Correspondence: loredana.frasca@iss.it; Tel.: +39-49906596-7
†   These authors contributed equally to this work.

**Abstract:** Systemic sclerosis (SSc) is a rare autoimmune disease whose molecular mechanisms are not yet fully understood. There is no definitive cure, and the main causes of death are pulmonary fibrosis and pulmonary arterial hypertension. Here, we focus on the interferon regulators factor 8 (IRF8), a factor involved in the type I interferon (IFN-I) signature, which is present in about half of SSc patients. Variants of this factor may play a role in autoimmunity, but little is known regarding the role of IRF8 in SSc pathogenesis. We carried out a literature search to address the association between the IRF8 factor and SSc susceptibility and clinical manifestations. The current studies appear to confirm a possible association between the alteration of the gene for IRF8 and SSc susceptibility. A link between IRF8 mutations and expression of a pro-fibrotic phenotype at the cellular level also emerges. Additional investigations are needed to confirm the role of IRF8 in SSc. However, IRF8 is worth consideration as a possible new disease marker of fibrosis in SSc patients.

**Keywords:** systemic sclerosis (SSc); interferon regulatory factor 8 (IRF8); diffuse cutaneous (dcSSc); limited cutaneous (lcSSc); interferon-I (INF-I); single nucleotide polymorphism (SNPs)

## 1. Introduction

SSc (Systemic sclerosis) or Scleroderma, is a systemic autoimmune disease of connective tissue, characterized by organ and skin fibrosis (due to dysfunction of fibroblast activity), vascular abnormalities or damage, autoimmunity, and excessive deposition of extra cellular matrix (ECM) [1,2]. SSc is considered as an orphan disease; there is no effective treatment to arrest the disease, and it represents the most serious connective tissue disorder. The incidence of SSc is 30 to 300 cases per million people, or 0.2 to 2.2 cases per million people per year; thus, it is considered a rare disease [3]. The main causes of death are pulmonary arterial hypertension and interstitial lung disease (ILD), and the treatment for skin and lung fibrosis involves immune suppression. Currently, the etiology remains unknown, due to the difficulty of diagnosis, rarity of the disease, and heterogeneity of clinical manifestations [4,5].

### 1.1. Genetics of SSc

SSc is a multifactorial disease with a complex etiology, in which both environmental and genetic factors are involved, like most autoimmune diseases. Arnett and colleagues in 2001, in an epidemiological study, identified the genetic base of SSc. They showed that incidence of SSc development was higher in the close relatives of those affected by SSc [6]. Since inheritance seems to be the strongest risk factor for developing SSc [7,8], several studies (gene association studies and genome wide analyses) were conducted to try to identify the genes responsible for this inheritance [9]. Currently, several susceptibility genes for SSc are known, such as *STAT3*, *STAT4*, *IRF5*, *IRF8*, *TNFSF4*, *CD226*, *CD247*, *MIF*,

*IL23R, IL2RA, IL-21, TLR2, CD226, BANK1, IRAK1, NFk, CCN2, TIMP1, AIF*, and *ACE* [6–9]. All are involved in the regulation of the immune functions or have effects on fibroblasts and endothelial cells activity. Recently, gene aberrations implicated in the development of the disease, such as SNPs (Single Nucleotide Polymorphisms), have been identified. In several cases, these SNPs are associated with noncoding regions (introns), responsible for gene regulation [10].

## 1.2. Manifestation of SSc and Subtypes

Some SSc manifestations are renal crisis, heart inflammation, pulmonary arterial hypertension, digital ulcers, and gastric reflux, mostly associated with high mortality [11,12]. In general, we distinguish two major temporally bound manifestations, early SSc (disease duration <5 years) and late SSc (duration >5 years). The disease is specifically characterized by the extent of skin involvement: limited cutaneous SSc (lcSSc) involves fibrosis of the skin distal to the elbows and/or knees but without truncal involvement, although skin thickening might occur on the face and neck; diffuse cutaneous SSc (dcSSc) involves skin both distal and proximal to the knees and/or elbows and/or truncal and involves internal organs. LcSSc patients show a low rate of disease progression, whereas dsSSc patients display a high mortality rate [13]. Nearly all patients show the classic clinical Raynaud's phenomenon before disease onset [14], and half of them have digital ulcers [15]. The actual diagnosis is made by fulfilling the EULAR classification criteria, even though not all patients can be classified into those criteria [16,17]. Furthermore, in 1962, a separate subset of SSc named SSc *sine scleroderma* (ssSSc) was described [18], characterized by peripheral vascular system, gastrointestinal system, lung, and heart involvement and by a lack of skin involvement [19], making diagnosis very difficult for the clinicians [20].

Searching biomarkers that allow one to infer prognosis and to distinguish different disease subtypes may lead to better management of the disease. Currently, anti-nuclear antibodies (ANAs) are used to identify the SSc subtypes. Typical clinical biomarkers are anti-centromere antibodies (ACAs), (elevated in lcSSc) [21], anti-topoisomerase I anti-bodies (ATAs) (elevated in dcSSc) [22], and anti-RNApolymerase-III antibodies (elevated in dcSSc) [23]. Other antibody specificities exist; however, they are less specific, like antifibrillarin antibodies (U3-RNP/Fibrillarin) [24], and anti-RNApolymerase-I and anti-RNApolymerase-II antibodies [8]. Their specificity and sensitivity varies depending on the autoantigen, type of immunoassay used, ethnicity, region, sex, etc.

## 1.3. Type I Interferon in SSc

SSc is characterized by a type I interferon (IFN-I) signature in about half of the patients, which is associated with worse prognosis, if present at early stages [25]. Studies suggested that IFN-I blockade improved the skin thickening score in early SSc, where the IFN-I-gene signature is associated with disease severity [26]. Furthermore, the induction or aggravation of SSc has been described in patients undergoing IFN-I-based therapies [27]. It has been shown that typical SSc autoantibodies activate innate immune cells, can supply inflammation, and contribute to the IFN-I signature and fibrosis. In addition, sera from SSc patients, mixed with apoptotic and necrotic material, induced IFN-α production in plasmacytoid dendritic cells (pDCs), apparently activated by immunoglobulin (Ig) immune complexes (ICs) formed by SSc autoantibodies [28,29]. It is now clear in the literature that there is involvement regarding self–nucleic acid recognition in inflammatory and autoimmune diseases though Toll-like receptors (TLRs), especially TLR7 and TLR9, which can induce type I IFN from pDCs but also promote the activation of self-reactive B cells [30,31]. In 2018, Ah Kioon and colleagues proposed a model where the aberrant expression of TLR8 on pDCs from SSc patients induces the secretion of chemokine (C–X–C motif) ligand 4 (CXCL4) (also known as platelet factor 4, PF4), which in turn can favor TLR responses, emerging as a dominant TLR driving fibrosis and promoting autoimmunity in SSc [32]. It is presently unclear from the scientific literature whether the signaling of TLR8 is also influenced by IRF8. CXCL4 is then found to form complexes with self-DNA in

the circulation and tissue of SSc patients and correlated with IFN-α in the blood or with interferon gene *Mx1* (MX dynamin like GTPase 1) in the skin [33].

CXCL4 can form complexes also with the RNA and complexes formed by CXCL4 and "self" RNA are also able to activate human pDCs to produce IFN-I [34]. Therefore, molecules responsible for IFN-I induction or involved in IFN-I signaling are worth being studied and may also represent useful biomarkers for SSc. Furthermore, new therapeutic agents should be tested for the capacity to block the IFN-I pathways, CXCL4, and pDCs [34].

### 1.4. IRF Family in Humans and Mice

A class of transcription factors called interferon regulatory factors (IRFs) are involved in IFN-I signaling. These are key mediators of immune responses in the host, immunomodulation, and differentiation of hematopoietic cells [35]. In 1988, two groups independently identified the first IRF, called IRF1 [36,37]. Currently, the IRFs characterized in humans and mice are depicted in Figure 1. As highlighted in the figure, IRF8 specifically enters into the signaling of TLR7 and TLR9. These are important TLRs implicated in various autoimmune disorders (see below). In humans, the following functional subgroups have been identified: IRF1 and IRF2 (promoting T-helper 1 (Th1) responses), IRF3 and IRF7 (involved in antibacterial and antiviral innate immunity), IRF4 and IRF8 (expressed only by myeloid and lymphoid lineages of the immune system, involved in B-cell development and T helper cell differentiation) [38], IRF5 and IRF6 (with pro-inflammatory role and regulatory effects on apoptosis), IRF9 (as part of the Interferon-stimulated gene factor 3, and ISGF3, a trimeric complex formed with STAT1 and STAT2) [39]. The main function of IRFs is to positively or negatively regulate the transcription and the consequent expression of IFN-I. Based on the composition and structure of the C-terminal region of the IRF proteins, IRF1, IRF3, IRF5, IRF7, and IRF9 were described as activators, whilst IRF2 and IRF8 as repressors [35]. In addition, it has been observed that the IRF family members can interact with the STAT protein family members, NFkB and PU-1, thereby activating and regulating a broad spectrum of genes [40]. All members of the IRF family possess a similar structure, with a DNA binding domain (DBD) at the N-terminus and a helix–turn–helix motif. This domain is essential for the binding to the consensus sequence, IFN-stimulated response element (ISRE), which coordinates the expression of IFN-I [41,42]. Importantly, it has been shown that the IRFs play a role in the pathogenesis of different autoimmune diseases: *IRF1* polymorphisms have been identified in patients with juvenile idiopathic arthritis (JIA) [43], Behcet's disease (BD) [44], in multiple sclerosis (MS) [45], and in rheumatoid arthritis (RA) [46]; *IRF2* and *IRF3* genetic variants are found to be associated with susceptibility to Systemic lupus erythematous (SLE), an autoimmune systemic disease characterized by an aberrant IFN-I signature [47,48]. *IRF5* risk variants have been reported, again, in patients with SLE [49]. The same type of variants are reported in RA [50]. According to another study, IRF5 variants are also implicated in multiple sclerosis (MS), an autoimmune disease affecting the brain, where an aberrant IFN-I signature can be apparent in a disease subgroup [51]. Finally, *IRF7* and *IRF8* polymorphisms are also associated with a risk of SLE [52,53]

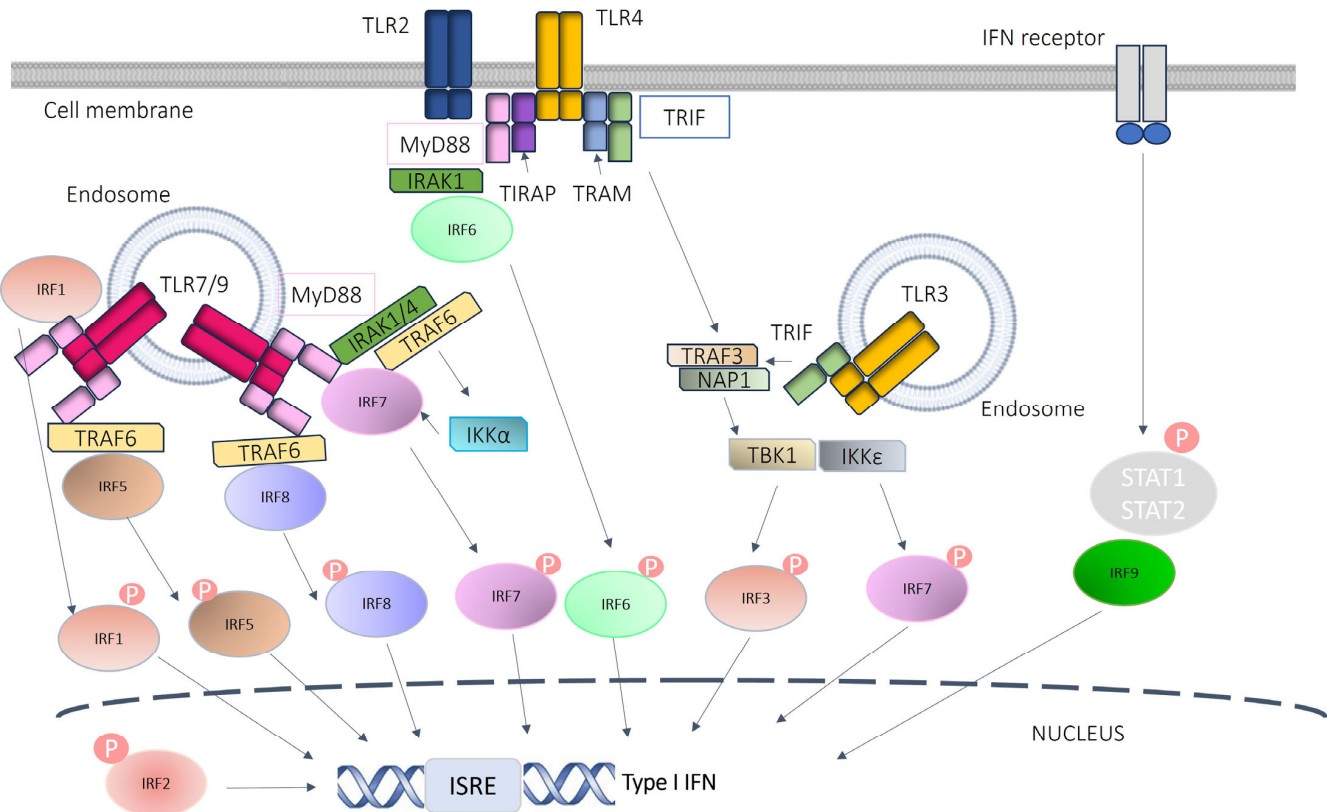

**Figure 1.** Different IRF molecules control nucleic acid-sensing TLR signaling. IRF8, highlighted in the red square, is working thought the endosomic TLR7 and TLR9. It is still unclear the role of TLR8. IKK, inhibitor of nuclear factor kappa-B kinase; IRAK, interleukin-1 receptor-associated kinase; ISRE, IFN-stimulated response element; MyD88, myeloid differentiation primary response protein 88; STAT, signal transducer and activator of transcription; TLR, Toll-like receptor; TRAF, Tumor necrosis factor (TNF) receptor-associated factor; TRAM, Toll-receptor-associated molecule; TRIF, TIR domain-containing adaptor–inducing IFN. Adapted from Zhang et al., 2015 [54], and Negishi et al., 2018 [55].

### 1.5. IRF Genes in SSc

The *IRF5* was the first gene identified to be associated with SSc [56,57]. In 2013, Sharif and colleagues showed that an SNP in the IRF5 promoter region (rs4728142), associated with lower IRF5 transcript expression, was predictive of longer survival and milder ILD in SSc patients [58]. Also, other members of IRF family genes have been linked to SSc, including *IRF1* [59], *IRF4* [60,61], *IRF7* [60,62], and *IRF8* [63,64]. SNP of the IRF1 locus (rs2548998) acted as an eQTL (expression quantitative trait loci) for IRF1 expression [59]. Carmona et al., found an association of IRF7 with the presence of ACA in SSc patients, so this locus may represent a risk factor for autoantibody production in SSc [62]. Microarray studies also described up-regulation of IRF7 mRNA level in PBMCs of SSc patients with early diseases [65], but another study in 2011 did not confirm the previous result [66]. The association of *IRF8* genetic variants with SSc described in the literature supports an involvement of dendritic cells and B cells in the development of SSc.

### 1.6. Role of IRF8 in Autoimmune Diseases

IRF8, interferon regulatory factor 8, originally termed interferon consensus sequence binding protein (ICSBP), is a 50 kDa protein that negatively regulates IFN-I. The expression of IRF8 has long been thought to be restricted to hematopoietic cells since it is abundantly expressed particularly in B cells and DC subsets [67]. Recently, analysis of IRF8 expression by single cell RNA sequencing showed that IRF8 is also expressed in several non-hematopoietic cells such as enterocytes, melanocytes, intestinal cells, and in epithe-

lial cells in the skin, liver, lung, and heart [68]. In hematopoietic cells, IRF8 is involved in myeloid cells differentiation and monocytes and macrophages development and the acquisition of effector functions [69]. IRF8 also negatively regulates osteoclast differentiation, and indeed IRF8 knock-out mice show a reduction in bone development [70]. IRF8 deficiency in humans results in a severe immunodeficiency, as characterized by susceptibility to infections due to a loss of DC subsets, CD14$^+$ and CD16$^+$ monocytes, and a decreased level of NK cells with reduced activity [71]. Indeed, IRF8 seems essential for myeloid cell lineage commitment and differentiation in mice and humans, since its deletion leads to huge accumulation of CD11b$^+$Gr1$^+$ immature myeloid cells (IMCs). Currently, genetic variants of IRF8 have emerged, which correlate with increased susceptibility to SLE and MS [72]. For example, a risk variant for SLE, rs2280381, is found in the area that regulates IRF8 expression by spatially interacting with the promoter of IRF8, influencing the methylation status and, consequently, its expression [53,73]. It has also been observed that IRF8, in association with IFN-I, can negatively regulate or repress the expression of BAFF, a cytokine crucial for the development and selection of B cells. Notably, BAFF has recently been identified as a new target for the treatment of SLE [74]. IRF8 promotes B cell differentiation and DC development [75]. Moreover, IRF8, by physically interacting with TRAF6, modulates the TLR-guided signaling pathway and appears to contribute significantly to the crosstalk between IFN-γ signaling pathways and TLRs, working as a link between innate and adaptive immune responses [76]. In 2015, GWASs were described to have 16 genetic variants significantly associated with autoimmune disorders in the IRF8 locus [77]. In addition, the CRISPR array, a high-throughput screen for the disease-related susceptibility loci, has had a genomic region, RefSNP (rs) 2280381, described which is located 64 kb downstream of IRF8 that act as an enhancer, which regulates IRF8 expression in monocytes [78].

### 1.7. IRF8 in SSc

Few articles on IRF8 polymorphisms show that this factor is either positively or negatively associated with SSc. These studies may also explain the contribution of B cells, dendritic cells and macrophages to SSc pathogenesis and shed light on molecular mechanisms involved in the fibrotic process. Given the recent interest in these genetic mechanisms and in the family of IRF factors, we present here (next paragraphs) evidence that suggests IRF8 as a possible determinant in SSc development and pathogenesis, with the goal to stimulate future and more in-depth analyses of IRF8's role in SSc.

## 2. Specific Role of IRF8 in SSc

A narrative search was conducted in the scientific literature (PRISMA protocol not followed), on PubMed and Google Scholar, searching for the following keywords "IRF8 and SSc", "IRF8 polymorphisms in the SSc", the "effect of IRF8 regulation in SSc", "IRF8 and biomarker and SSc", "IRF8 regulation in macrophage in SSc". No time limit was placed on the research performed. We included epidemiological studies, animal models, and in vitro cultures.

### 2.1. IRF8-KLF4 (Kruppel-like Factor 4) Interaction

Krüppel-like factor 4 (KLF4) is a transcription factor that regulates several cellular processes such as proliferation, cell growth, and differentiation. It can interact with other factors, modulating the efficiency to bind the DNA [79]. In 2013, Kurotaki and colleagues discovered an essential role of IRF8-KLF4 transcription factor cascade in murine monocyte differentiation. Basically, by combining chromatin immunoprecipitation sequencing with gene expression profiling, they have shown that IRF8 works as a key regulator of the development of several DC subsets, also indicating a crucial role in the mononuclear phagocyte system [80].

## 2.2. GWAS (Genome-Wide Association Study)

Recently, GWASs have proven to be a very promising tool in the identification of thousands of genetic variants associated with the pathogenesis of complex diseases [81] and in the discovery of new and more tailored drugs [82].

In 2013, a GWAS conducted by Terao and colleagues included *IRF8* as a susceptibility gene for SSc in a Japanese population, particularly in the lcSSc subgroup [83]. In addition, SSc seems to share many susceptibility genes with RA, including *IRF8*, so it would be interesting to transfer the analysis from a Japanese to European population. In the meta-GWAS, conducted by Lòpez-Isac and colleagues, *IRF8* was one of the twenty-seven signals associated with SSc. RefSNP (rs) 11117420, a locus of 40 kilobases of the *IRF8* transcription starting site, has been shown to interact with and regulate its own promoter [61,81].

## 2.3. IRF8 Expression in SSc Subtypes

It has been observed that certain SSc clinical features and the presence of disease specific auto-antibodies are variable in different countries and ethnicities [84]. This supports the fact that genetic factors may influence the clinical features of SSc and the type of auto-antibodies [85]. In addition, there is evidence indicating that genetic factors can be also helpful in the early diagnosis of specific clinical subtypes of SSc. In 2011, Gorlova's group showed a strong association between the *IRF8* gene and the lcSSc subtype and the ACA positive subgroup [63], suggesting that a genetic heterogeneity determines the clinical manifestation and, in particular, the autoantibody subtypes of SSc. These findings may prompt reconsideration of the current classification of SSc patients, leading to novel therapeutic targets for this devastating autoimmune disease.

## 2.4. IRF8 in SSc Fibrosis

Starting from the study by Guo et al. published in 2017, where the authors have shown that the inhibition of IRF8 negatively affected the M1 macrophage polarization, delaying the wound healing [86], recently another group investigated IRF8 levels in SSc monocytes as well as IRF8 regulation in monocytes and macrophages in SSc fibrosis [87]. In particular, the authors have demonstrated that mRNA levels of *IRF8* were reduced when measured in PBMCs of dcSSc patients, as compared to cells of lcSSc patients and healthy controls. The *IRF8* gene was also downregulated when measured in SSc circulating monocytes. In addition, it seemed that IRF8 had a strong impact on the pathogenesis of skin sclerosis, since the reduction in this factor, in terms of mRNA and protein, inversely correlated with the Rodnan total skin thickness score. In 2017, Qiu and colleagues demonstrated that IRF8 demethylation can act as suppressor of Th1 response, via IL-6 regulation. So, IRF8 downregulation in circulating monocytes may cause a shift from Th1 to Th2, which is important for establishing SSc symptoms [88]. All these observations strongly suggest that IRF8 plays a key role in the effector functions of monocytes during the skin fibrosis process. Furthermore, following IRF8 gene silencing, monocytes tended to preferentially differentiate into a profibrotic phenotype as they show a high mRNA expression for profibrotic cytokines and chemokines. Of note, macrophages derived from monocytes are of two types, M1 and M2, which have key roles in SSc. The silencing of IRF8 in monocytes induces their differentiation into the M2 type, although more in-depth analyses show a mixed M1 and M2 phenotype, since both protein subtypes are detected (TNF-a and IL-6), (MCP-1, EGR1, TGFβ, and αSMA). This agrees with other studies, which reported that IRF8 knockout mice reduced M1-specific gene expression and established a preferential environment for M2 subtype activation [89]. Other studies have found that IRF8 is required for M1 polarization induced by the Notch-RBP-J pathway, leading to a switch to the IRF8 down-regulated M2 phenotype [90]. At the same time, monocytes also displayed high levels of CEBPB (CCAAT/enhancer-binding protein beta), a transcription factor crucial in the formation of an M2 macrophage phenotype. Therefore, it can be hypothesized that siIRF8 monocytes may have pro-fibrotic characteristics with segregated nuclei containing atypical monocytes. It is unclear which conditions might influence this phenotype in vivo

and how these cells are relevant in the maintenance of fibrosis at wound sites, but it is believed that IRF8 down-regulation is involved. From the reported data, an increased number of infiltrating macrophages was observed in the skin of IRF8 KO mice compared to controls, which had massive expression of pro-fibrotic markers such as COL1A1 (collagen type I alpha 1 chain) and αSMA (Smooth muscle alpha-actin). This is probably due to the involvement of MCP-1 (Monocyte chemoattractant protein-1), an important factor in the progression and onset of systemic sclerosis, which promotes collagen experience in fibroblasts and is up-regulated in patients with SSc [91,92].

### 2.5. IRF8 SNPs and SSc Susceptibility

As mentioned above, some of the susceptibility genes for SSc are similar to other autoimmune diseases. Therefore, Arismendi and colleagues in 2015 started their investigation on 16 SNPs published at that time as susceptibility factors for SSc and SSc subtypes. They showed for the first time the specific association between *IRF8* rs11117432 SNP and SSc susceptibility [64]. This association seemed to be more present in SSc patients with ACA antibodies and in those with lcSSc, suggesting a role in the pathogenesis of SSc at least in these groups of patients. As already documented, IRF8 is a nuclear protein that, upon activation of pathogen-associated molecular agents (PAMPs), moves into the cytoplasm, activating the NF-κB and TLR signaling pathways. IRF8 and NFKB gene variants can interact, through an epistatic interaction, and have a role in determining patients' susceptibility to SSc [64]. *IRF8* rs11117432 might have a role in the regulation of extracellular matrix and collagen deposition in fibrotic disease by the modulation of inflammation (pro-inflammation vs. anti-inflammation). More recently, in 2023, by generating promoter capture Hi-C data for CD4 T cells and CD14 monocytes from a small cohort of SSc patients, it was shown that the *IRF8* rs11117420 variant can be a new hypothetical SNP for SSc [59].

### 2.6. Physical Interaction with Chromatin for IRF8 Expression in Monocytes

Studies in mouse models have observed that the deletion of an enhancer region leads to decreased IRF8 gene expression, leading to an overproduction of Ly6C$^+$ inflammatory monocytes [92]. In this case, the most plausible hypothesis is that the variant of IRF8 is associated with SSc through a possible physical interaction with chromatin in CD14$^+$ monocytes. All this leads to an alteration in genetic expression, thus inducing genetic imbalance at the cellular level.

Figure 2 depicts the possible effect of IRF8 down-regulation in SSc monocytes. At one side, IRF8 down-regulation favors M2 development, which is involved in SSc. However, a second general effect of IRF8 down-regulation is the lack of regulation of IFN-I, as IRF8 acts as a negative regulator of the IFN-I signaling (also depicted in Figure 2). However, the effect in SSc of IRF8 mutation should be studied in more detail. For example, a study in a mouse model, although not related to SSc, has shown that a single-point mutation in the IRF8 gene, the Irf8R294C mutation, was able to strongly impair the IFN-I-mediated response by murine pDCs [93]. Thus, although IRF8 is acting as a negative regulator of the IFN-I pathways, some changes in IRF8 could also inhibit IFN-I by the most important cells that produce IFN-I, the pDCs. This paper is interesting as pDCs are strongly involved in SSc pathogenesis. The latter assumption is derived from several pieces of evidence, as described above. The authors of Ref. [93] suggest the utility of this mouse model harboring the IRF8 R294C mutation as a tool to investigate the effect of IRF8 in autoimmune diseases characterized by an aberrant IFN-I signature. Among these IFN-I dominated diseases, SSc and SLE are the most relevant.

In Table 1, we have summarized the most important papers (in chronological order) that have studied IRF8 involvement in SSc, as well as the model systems used to reach their conclusions. We have evidenced the main results of each study.

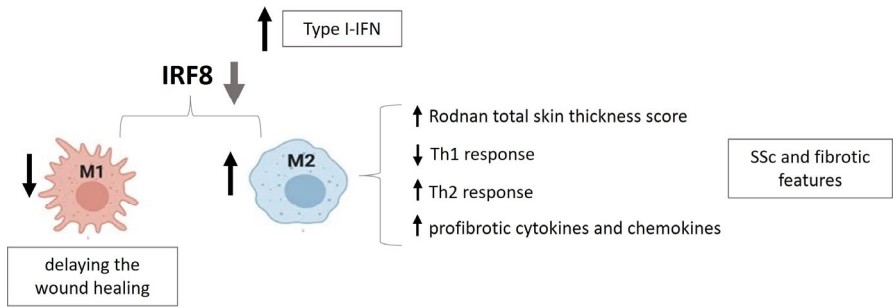

**Figure 2.** Model for IRF8 downregulation effects in macrophages leading to systemic sclerosis. M1: M1 macrophages; M2: M2 macrophages.

**Table 1.** IRF8 involvement in SSc.

| Authors | Years of Publication | Methods | Results |
| --- | --- | --- | --- |
| Gorlova et al. [63] | 2011 | Genome-wide association study | Association between the *IRF8* gene and the lcSSc subtype and the ACA positive subgroup |
| Kurotaki et al. [80] | 2013 | Chromatin immunoprecipitation sequencing; gene expression | IRF8-KLF4 in monocyte differentiation |
| Terao et al. [83] | 2013 | Genome-wide association study | *IRF8* as susceptibility gene for SSc in a Japanese population |
| Mahoney et al. [91] | 2015 | Microarray | Connection between inflammatory- and fibroproliferative-specific genes |
| Arismendi et al. [64] | 2015 | Gene expression, genotyping | IRF8 rs11117432 SNP and SSc susceptibility in lcSSc subtype and the ACA positive subgroup |
| Qiu et al. [94] | 2017 | Gene expression, quantitative methylation analysis | IRF8 demethylation as suppressor of Th1 response |
| Lòpez-Isac et al. [81] | 2019 | Genome-wide association study | IRF8 rs11117420 SNP interaction and regulation of its promoter |
| Ototake et al. [87] | 2021 | Gene expression, immunoblotting, Irf8 knockout mice | Involvement of altered IRF8 regulation in monocytes and macrophages in lcSSc |
| González et al. [59] | 2023 | Promoter capture Hi-C | IRF8 rs11117420 variant as new hypothetical SNP for SSc |

## 3. Conclusions

As we reviewed here, SSc is a heterogeneous disease, and in recent years, the scientific focus has shifted to the identification of novel mutated or deleted gene loci that may explain the presence or absence of a certain cellular phenotype or molecular phenomena. The identification of mutated, non-mutated, or deleted genes for IRF8 can currently be a new area of research in SSc. Indeed, the existing articles on this subject confirm IRF8's involvement in SSc (particularly in lcSSc and in ACA positive patients). The dysfunction of the IRF8 factor is found to be specific for macrophages and monocytes in SSc, which are cell types crucial in SSc pathology. In the literature, in different GWASs, the presence of SNPs at the IRF8 gene locus is reported.

Future studies focused on the possible role played by this transcription factor in SSc development and maintenance, and on the possible role of IRF8 in other cell types in SSc, may be crucial for a better understanding of the role of IRF8 in SSc pathogenesis and IRF8's role as a new possible biomarker for SSc. For this kind of analysis, more studies, not directly linked to the SSc fields, could be illuminating, as some IRF8 mutations have been found as rare events in particular cases in patients. In this regard, there is a study [95] which describes a mutation of IRF8 called IRF8R291Q in one patient. This mutation is orthologous to the murine IRF8R294C mutation. In the clinical case described, this mutation has several immunological consequences, which culminate in recurrent viral infections experienced by the patient and is linked to a deficit of the immune cell

functions or to wrong pathways of immune cell activation and aberrant expression of functional molecules. Excessive granulopoiesis was evident with a massive neutrophil infiltration in the lungs of the studied subject. Many mediators of inflammation were up-regulated, and among these were Tumor necrosis factor (TNF)-α, IL-8, IL-6, IL-1β, and several chemokines [95]. Impairment of B-cells, T-cells, and natural killer (NK) cells was also observed by the authors. These results clearly indicate that additional cell types are affected by IRF8 mutations. For instance, T-cells were affected, so they did not express C-X-C motif chemokine receptor 3 (CXCR3), an important factor that drives immune cells into inflamed tissues. The authors also observed profound defects in T-helper (TH)1, TH17, and CD8 effector memory development. Thus, a single-point mutation in IRF8 seems to have a great effect on many immune cell types. The paper describes biallelic mutations in one person, which affect all these immune cells. An additional paper reports that IRF8 can control Th1 immune responses independently from the T-box transcription factor TBX21, also called T-bet, a factor crucial for TH1 polarization. IRF8 was also found to control T regulatory cells (Tregs). Indeed, the authors reported that expression of Forkhead Box P3 (Foxp3), the specific factor for Tregs, induced IRF8 in the Tregs [96]. From both papers cited above, we can conclude that IRF8 is involved in a T-cell phenotype that has relevance also for SSc. Indeed, there are several studies that addressed the role of T-helper cells in SSc and the fact that Tregs cells do not properly work in SSc [97,98]. Finally, a study showed that circulating CD123⁺CD127⁺ lymphoid progenitors sustain human innate lymphoid cells (ILCs) in addition to T-cell development. These progenitors also reside in the thymus and have an increased expression of IRF8 and CD123. The authors utilize CD123 expression as a surrogate for IRF8 expression in these cells [99]. Although only a few studies have examined the role of ILCs in the pathophysiology of human SSc, some studies have analyzed these cells in SSc [100]. All these studies suggest that the role of IRF8 is pleiotropic as it affects several cell types that can have an impact on SSc pathogenesis. Future research in these directions is necessary in the SSc field.

**Author Contributions:** Writing—first draft preparation, G.O.; Writing—review and editing, A.M. with L.F. and G.O.; Conception, K.S.; Literature search A.M., L.F. and G.O.; Supervision the writing of the final revision, L.F.; Preparation of Figures and Tables, A.M. All authors have read and agreed to the published version of the manuscript.

**Funding:** FOREUM research grant to L.F.

**Institutional Review Board Statement:** Not applicable.

**Informed Consent Statement:** Not applicable.

**Conflicts of Interest:** The authors declare no conflict of interest.

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
