# Peer review of "The Role of IRF8 Polymorphisms in Systemic Sclerosis Development and Pathogenesis"

_jmp, doi:10.3390/jmp5010008_

Round 1
Reviewer 1 Report
Comments and Suggestions for Authors
In this manuscript the authors made a review of IFR8 role in SSc. My first comment is to send a big congratulation to the authors. The manuscript is very well done in my opinion and is very useful to refresh the IFN role in SSc. I only have a minor comment:
Inside introduction (ROW 32 page 1): in phrase " The main causes of death are pulmonary arterial hypertension and fibrosis" i suggest to change by " the main causes of death are pulmonary arterial hypertension and INTERSTITIAL LUNG DISEASE" . To use fibrosis here is not clear for clinicians at least.
No more comments on my side. Congratulations again
Author Response
Response to reviewer:
We thank the reviewer for the appreciation of our work.
We have done the correction suggested in the Introduction:
"The main causes of death are pulmonary arterial hypertension and interstitial lung disease (ILD), and the treatment for skin and lung fibrosis involves immune suppression".
Some changes have been added and one figure, Figure 1, and all changes are underlined. There are some additions to the Conclusions.
Loredana Frasca (for all authors)
Reviewer 2 Report
Comments and Suggestions for Authors
The topic, IRF8 in Systemic Sclerosis is current and may interest the Journal of Molecular Pathology’s readers. The introduction of interferon inhibitor treatments has made this topic of particular interest to rheumatologists and other physicians. This review helps to understand one of the main pathological processes of systemic sclerosis. There is only one explanatory figure of the processes in the manuscript, which can help to better understand the complicate relationships. The main content of the review is well written and understandable.
However, there are too many abbreviations, and the meaning of abbreviations is not clarified in some places. eg.: pDCs, PBMCs.
Not much practical value for practicing physicians but helps and supports causes in gaining attention and in understanding the pathogenesis of SSc.
Author Response
We thank the reviewer for the careful evaluation or our review.
We have corrected and explained the abbreviations where needed.
Some changes have been added, and one figure, Figure 1, and all changes are underlined.
There are some additions to the Conclusions.
Loredana Frasca (for all authors).
Reviewer 3 Report
Comments and Suggestions for Authors
Interesting and detailed data collection from existing Literature with some propositive ideas for future research.
In the Introduction, Authors distinguish two different subtypes of SSc, without mention of Systemic sclerosis sine scleroderma ( De Angelis et al, RMD Open 2023), a quite rare variant of the disease. It could be useful to cite this serological feature.
In all the text, some acronym need to be explicated before the use.
In line 98 (IRF family in humans and mice) I can read: In humans IRFs can be divided into five functional subgroups..., but only four groups are listed, please check and rectify if needed.
In Table 1, is not clear the criterion opted to list the papers. I suggest to take an option (year of publication? Methods?) to present data.
The title choosen is not quite appealing, I suggest the Author to try to give more character.
Author Response
Interesting and detailed data collection from existing Literature with some propositive ideas for future research.
We thank the reviewer for the appreciation of our work. We also added some new comments in the conclusions and additional references. All changes are underlined.
In the Introduction, Authors distinguish two different subtypes of SSc, without mention of Systemic sclerosis sine scleroderma ( De Angelis et al, RMD Open 2023), a quite rare variant of the disease. It could be useful to cite this serological feature.
We thank the reviewer for the careful review of our manuscript.
We agree that reference to the SSc sine scleroderma is lacking and should be added. We thank for this suggestion.
In all the text, some acronym need to be explicated before the use.
We have explicated the acronyms and abbreviations.
In line 98 (IRF family in humans and mice) I can read: In humans IRFs can be divided into five functional subgroups..., but only four groups are listed, please check and rectify if needed.
We thank the review for this correction. We have changed the paragraph (che changes are underlined) and we also added a Figure on IRFs (Figure 1). We hope that this makes the paragraph more clear.
In Table 1, is not clear the criterion opted to list the papers. I suggest to take an option (year of publication? Methods?) to present data.
We put the papers in chronologic order in the Table 1.
The title chosen is not quite appealing, I suggest the Author to try to give more character.
We changed the title according to the reviewer suggestion:
“The role of IRF8 polymorphisms in Systemic Sclerosis development and pathogenesis”
Loredana Frasca (for all authors)
